# GRADIENT-BASED COUNTERFACTUAL EXPLANATIONS USING TRACTABLE PROBABILISTIC MODELS

## ABSTRACT

Counterfactual examples are an appealing class of post-hoc explanations for machine learning models. Given input $x$ of class $y_1$, its counterfactual is a contrastive example $x'$ of another class $y_0$. Current approaches primarily solve this task by a complex optimization: define an objective function based on the loss of the counterfactual outcome $y_0$ with hard or soft constraints, then optimize this function as a black-box. This "deep learning" approach, however, is rather slow, sometimes tricky, and may result in unrealistic counterfactual examples. In this work, we propose a novel approach to deal with these problems using only two gradient computations based on tractable probabilistic models. First, we compute an unconstrained counterfactual $u$ of $x$ to induce the counterfactual outcome $y_0$. Then, we adapt $u$ to higher density regions, resulting in $x'$. Empirical evidence demonstrates the dominant advantages of our approach.

## 1 INTRODUCTION

Explaining decisions made by intelligent systems, especially black-box models, is important. First of all, a model that can explain their decisions is more likely to gain human trust, especially if there are significant consequences for incorrect results, or the problem has not been sufficiently studied and validated in real-world applications (Simpson, 2007; Hoffman et al., 2013; Doshi-Velez & Kim, 2017). Secondly, explanations can promote fairness by exposing bias towards protected attributes in the system (Prates et al., 2019; Buolamwini & Gebru, 2018; Obermeyer et al., 2019). In addition, explanations are a helpful tool for debugging purpose (Holte, 1993; Freitas, 2014; Rudin, 2019).

Counterfactuals are closely related to classes of explanation, which much of the early work in AI on explaining the decisions made by an expert or rule-based systems focused on (Wachter et al., 2017). As the name implies, counterfactual is an alternative example counter to the fact which takes the form "If X had not occurred, Y would not have occurred". Explicitly establishing the connection between counterfactuals and explanations, Wachter et al. (2017) concluded that counterfactual explanations can bridge the gap between the interests of data subjects and data controllers that otherwise acts as a barrier to a legally binding right to explanation. They further formulate the problem of finding counterfactual explanations as an optimization task that minimizes the loss of a desired outcome w.r.t. an alternative input combined with regularization for distance to the query example. In a similar vein, the dominant majority of recent counterfactual explanation methods are proposed based on their adapted objective functions for optimization.

Unfortunately, these deep learning approaches have some downsides: First of all, they are very time-consuming for generating explanations due to the iterative optimization process, because for every optimization iteration the model needs to be evaluated at least once and it may take a lot of iterations until a candidate is found. Secondly, the objective function is usually highly non-convex, which makes it more tricky to find a satisfying solution in practice. Thirdly, the optimizer involves additional hyperparameters that need to be carefully selected, such as learning rate. Furthermore, although these methods optimize for the "closest possible world" by constraining the distance of the counterfactual to the query instance, they are still not aware of the underlying density and the data manifold. As a direct consequence, the counterfactuals often appear rather unnatural and unrealistic although they are as close as possible to the query instance. See figure 3 for example. This imposes difficulties for humans to understand the explanations. As Nickerson (1998) argues, humans exhibit confirmation bias, meaning that we tend to ignore information that is inconsistent with our prior

beliefs. In analogy, unrealistic counterfactuals also have very limited effect for communicating explanations to humans.

To improve upon these limitations, we propose a novel approach to generate counterfactual explanations. Unlike the overwhelmingly dominant deep learning approach, we decouple the goal of perturbing the prediction and maintaining an additional constraint on the perturbation. Specifically, we view the task as a two-step process, whereby the first step is to perturb a sample towards a desired class and the second step is to constrain the perturbed sample to be close to the data manifold so it is realistic and natural. The first goal aligns with adversarial attacks (Goodfellow et al., 2015; Brown et al., 2017; Yuan et al., 2019), where a small perturbation is computed to perturb the prediction. In the literature there are gradient-based methods (Kurakin et al., 2016; Moosavi-Dezfooli et al., 2016) and optimization-based methods (Szegedy et al., 2013; Carlini & Wagner, 2017). Exploiting the fact that gradient corresponds to the direction of steepest ascent, gradient-based perturbation is not only very easy to implement with current deep learning frameworks, but also much faster to compute compared to optimization-based perturbation. Therefore we adopt a gradient-based approach for the first step. To ensure the quality and effectiveness of the second step, we learn a sum-product-network (SPN) (Poon & Domingos, 2011), a tractable density model, on the input, and we perturb the example further based on the gradient that directs to steepest ascent of likelihood, evaluated on the SPN.

In summary, we make the following contributions:

1. We propose the first approach for generating counterfactual examples using probabilistic circuits to ensure density-awareness in a tractable way.

2. We experiment with complex and high-dimensional real-world dataset, which was not dealt with using domain-agnostic methods in the literature.

3. We give empirical evidence to demonstrate the advantages and effectiveness of our approach: visually appealing examples with high density, fast computation, and high success rate.

We proceed as follows: First, we give an overview of the literature and summarize the shortcomings of the existing approaches. Then we introduce RAT-SPNs, preparing for presenting our approach. Finally, we evaluate our approach via empirical evidence and conclude our work.

## 2 RELATED WORK ON COUNTERFACTUAL EXPLANATIONS

Research attention on counterfactual explanations has been increasingly raised since Wachter et al. (2017) presented the concept of unconditional counterfactual explanations as a novel type of explanation of automated decisions. Identifying the resemblance between counterfactual explanations and adversarial perturbations, Wachter et al. (2017) proposed to generate counterfactual explanations based on the optimisation techniques used in the adversarial perturbation literature. Specifically, a loss function is minimized w.r.t. its input, using standard gradient-based techniques, for a desired output and a regularizer penalizing the distance between the counterfactual and the query.

Following Wachter et al. (2017), Mothilal et al. (2020) augmented the loss with diversity constraint to encourage diverse solutions. However, these approaches do not have explicit knowledge of the underlying density or data manifold, which may lead to unrealistic counterfactual explanations. A bunch of methods counteract this issue by learning an auxiliary generative model to impose additional density constraints on the optimization process. As common choice for density approximator, Variational Autoencoder (VAE) (Kingma & Welling, 2014) and its variants (Klys et al., 2018; Ivanov et al., 2019) are used. For instance, Dhurandhar et al. 2018 proposed contrastive explanations method (CEM) for neural networks based on optimization. The objective function consists of a hinge-like loss function and the elastic net regularizer as well as an auxiliary VAE to evaluate the proximity to the data manifold. Ustun et al. 2019 defined the term recourse as the ability of a person to change the decision of a model by altering actionable input variables. Recourse is evaluated by solving an optimization problem. Joshi et al. 2019 provided an algorithm, called RE-VISE, to suggest a recourse, based on samples from the latent space of a VAE characterizing the data distribution. Pawelczyk et al. 2020 developed a framework, called C-CHVAE, to generate faithful counterfactuals. C-CHVAE trains a VAE and returns the closest counterfactual due to a

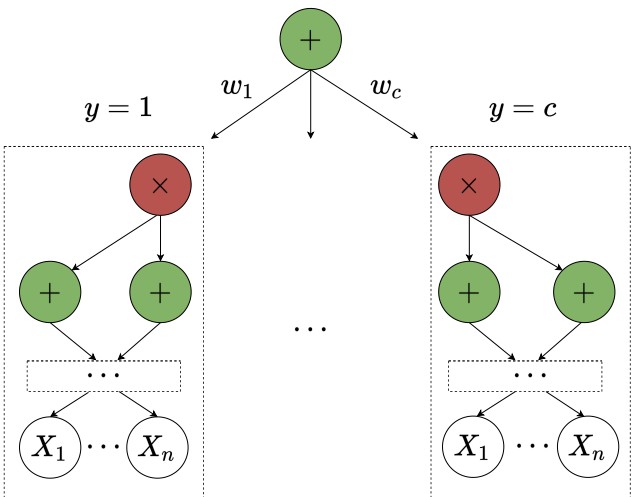

Figure 1: The SPN structure used to estimate a distribution as a mixture of class-conditional densities. Each class-conditional density is in turn represented by a sub-SPN. Classification is done via Bayes' rule.

nearest neighbour style search in the latent space. Downs et al. 2020 proposed another algorithmic recourse generation method, CRUDS, that generates multiple recourses satisfying underlying structure of the data as well as end-user specified constraints. Based on a VAE-variant, CRUDS uses a Conditional Subspace Variational Autoencoder (CSVAE) model (Klys et al., 2018) that is capable of extracting latent features that are relevant for prediction. Another method called CLUE Antorán et al. (2020) is proposed for interpreting uncertainty estimates from differentiable probabilistic models using counterfactual explanations, by searching in the latent space of a VAE with arbitrary conditioning (VAEAC) (Ivanov et al., 2019).

Poyiadzi et al. 2020 proposed FACE, a graph-based algorithm to generate counterfactuals that are coherent with the underlying data distribution by constructing a graph over all the candidate targets. Besides, several domain-specific approaches are also emerging Olson et al. (2021); Goyal et al. (2019); Chang et al. (2019).

The aforementioned techniques have the following shortcomings: 1. The deep learning-like approach (Wachter et al., 2017; Mothilal et al., 2020; Dhurandhar et al., 2018; Ustun et al., 2019; Joshi et al., 2019) is too slow to generate explanations on the fly due to the iterative optimization process, which in turn comes with additional tuning parameters. 2. Although highly expressive, neural density estimators such as VAEs are highly intractable, which makes explicit density constraint on counterfactual explanations infeasible (Dhurandhar et al., 2018; Ivanov et al., 2019; Klys et al., 2018; Joshi et al., 2019; Pawelczyk et al., 2020). In addition, they come with the overhead of latent representation, which is not necessarily in good quality when maximum likelihood training is used to learn them (Alemi et al., 2018; Dai & Wipf, 2019).

We improve upon the aforementioned shortcomings by proposing the first counterfactual method using tractable probabilistic circuits, specifically sum-product networks (SPNs) (Darwiche, 2003; Poon & Domingos, 2011).

## 3 GRADIENT-BASED COUNTERFACTUAL EXPLANATIONS USING TRACTABLE PROBABILISTIC MODELS

Before explaining our idea, we first introduce the tractable probabilistic models we use.

An sum-product network (SPN) $\mathcal{S}$ over $\mathbf{X}$ is a tractable probabilistic model for $P(\mathbf{X})$ based on a directed acyclic graph (DAG). This graph indicates computation for probabilistic inference and consists of three types of nodes: univariate leaf nodes, sum nodes, and product nodes. Let $ch(\cdot)$ denote the children of a node. A sum node $S$ is weighted sum of its children, i.e. $S = \sum_{N \in ch(S)} w_{S,N} N$ where the weights $w_{S,N}$ are non-negative and sum to 1, i.e. $w_{S,N} \geq 0, \sum_N w_{S,N} = 1$. Sum nodes can be viewed as mixtures of their child distributions. A product node $P$ is product of its children, i.e. $P = \prod_{N \in ch(P)} N$. The root node represents $P(\mathbf{X})$. Products are factorized distributions,

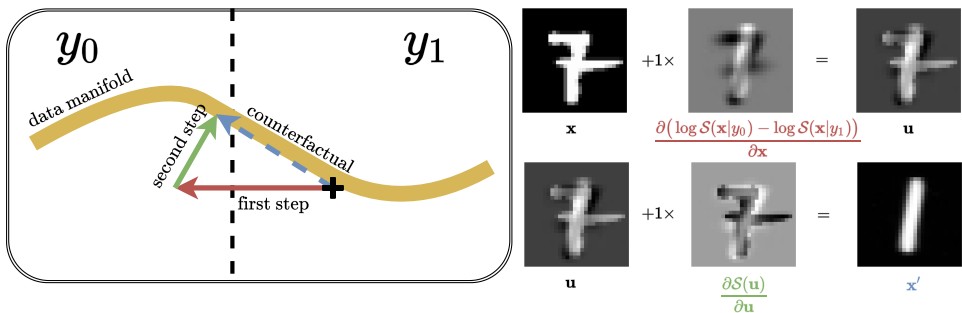

Figure 2: Left: Illustration of our gradient-based approach. Arrow indicates perturbation based on a gradient step. Right: An example on MNIST. The first row corresponds to the first gradient step for perturbing the prediction irregardless of the manifold. This perturbation removes some characteristics of the current class and adds some characteristics of the counterfactual class. The resulting sample **u** yields the desired class but obviously deviates from the training sets (It looks neither like a 1 nor 7). The second row corresponds to the second gradient step for generating in-distribution counterfactual by pushing the intermediate sample **u** to a region with higher density.

implying independence assumption among their children. SPNs allow for fast, exact inference on high-treewidth models.

Unlike most of the probabilistic deep learning approaches, SPNs permit exact and efficient inference. Specifically, they are able to compute any marginalization and conditioning query in time linear of the model's representation size. By employing SPNs for deep learning, *random and tensorized SPNs* (RAT-SPNs) (Peharz et al., 2020) are proposed using a simple approach to construct random SPN structure and combine it with GPU-based optimization. It is worth noting that RAT-SPNs are not fooled by certain out-of-domain image detection tests on which VAEs, normalizing flows (NFs), and auto-regressive density estimators (ARDEs) consistently fail (Choi & Jang, 2018; Nalisnick et al., 2018).

To learn the parameters $\omega$ of a given RAT-SPN structure $\mathcal{S}$ in generative setting to approximate a distribution $P^*(\mathbf{X})$, we assume i.i.d. samples $\mathcal{X} = \{\mathbf{x}_1, \ldots, \mathbf{x}_N\}$ are given. Then maximum likelihood estimation is employed, i.e. $\omega = \arg\max \frac{1}{N} \sum_{n=1}^{N} \log \mathcal{S}(\mathbf{x}_n)$, where $\mathcal{S}(\mathbf{x})$ is a distribution over $\mathbf{X}$ represented by the RAT-SPN $\mathcal{S}$.

Apart from the standard use for density estimation, RAT-SPNs can be used as a generative classifier as well. Consider a classification problem $f : \mathbb{R}^d \rightarrow \{1, \ldots, C\}$ with $C$ labels, $C$ roots are used to represent class-conditional densities $\mathcal{S}_c(\mathbf{X}) =: \mathcal{S}(\mathbf{X}|Y = y)$. The overall density distribution is then given by $\mathcal{S}(\mathbf{X}) = \sum_y \mathcal{S}(\mathbf{X}|y)P(y)$. See figure 1 for illustration. Bayes' rule is used to classify a sample $\mathbf{x}$: $\mathcal{S}(Y|\mathbf{x}) = \frac{\mathcal{S}(\mathbf{x}|Y)P(Y)}{\mathcal{S}(\mathbf{x})}$. In other words, a RAT-SPN of this special structure has dual use: It is both a density estimator $\mathcal{S}(\mathbf{X})$ and a classifier $\mathcal{S}(Y|\mathbf{X})$. We will use this model to demonstrate our approach because it can be used for classification and yield tractable density evaluation for free at the same time. However, our approach can be easily extended to deep neural classifiers by training an auxiliary RAT-SPN for density estimation. For more details on RAT-SPN, check out appendix.

Our approach is defined as two serial perturbations: In the first step, we maximize $\log \frac{\mathcal{S}(y'|\mathbf{x})}{\mathcal{S}(y|\mathbf{x})}$ to induce desired outcome $y'$, which is equivalent to maximizing $\log \frac{\mathcal{S}(\mathbf{x}|y')}{\mathcal{S}(\mathbf{x}|y)}$ since

$$\log \frac{\mathcal{S}(y'|\mathbf{x})}{\mathcal{S}(y|\mathbf{x})} = \log \Big( \frac{\mathcal{S}(\mathbf{x}|y')P(y')}{\mathcal{S}(\mathbf{x})} \frac{\mathcal{S}(\mathbf{x})}{\mathcal{S}(\mathbf{x}|y)P(y)} \Big) = \log \frac{\mathcal{S}(\mathbf{x}|y')}{\mathcal{S}(\mathbf{x}|y)},$$

when assuming a uniform class prior. Towards this end, we take a gradient step towards the steepest ascent of the counterfactual outcome. This results in the perturbed example $u$ where

$$\mathbf{u} = \mathbf{x} + \frac{\partial \big( \log \mathcal{S}(\mathbf{x}|y') - \log \mathcal{S}(\mathbf{x}|y) \big)}{\partial \mathbf{x}} * \epsilon_1.$$

This step is similar to generating an adversarial perturbation in that the sample $\mathbf{u}$ is expected to change the prediction to $y'$ from $y$ with a very slight change.

However, without additional constraints, the perturbed sample $\mathbf{u}$ is very likely to deviate from the underlying data manifold. In order to generate in-distribution counterfactuals, we maximize the density $P^*(\mathbf{u})$ of the current sample $\mathbf{u}$, which is approximated via the SPN $\mathcal{S}(\mathbf{u})$. In contrast to the density estimators used primarily in the literature, SPNs have the merit that they allow for tractable density evaluation. To maximize the density $\mathcal{S}(\mathbf{u})$, we take another gradient step towards its steepest ascent, i.e.

$$\mathbf{x}' = \mathbf{u} + \frac{\partial \mathcal{S}(\mathbf{u})}{\partial \mathbf{u}} * \epsilon_2.$$

$\mathbf{x}'$ is the final counterfactual explanation for $\mathbf{x}$ that yields $y'$ instead of $y$. See figure 2 for illustration.

## 4 EMPIRICAL EVALUATION

To illustrate the advantages of our approach, we designed experiments to evaluate it both qualitatively and quantitatively across several benchmark datasets. All the experiments are implemented in Python and Tensorflow, running on a Linux machine with two Intel Xeon processors with 56 hyper-threaded cores, 4 NVDIA GeForce GTX 1080 under Ubuntu Linux 14.04.

**Datasets**: We experiment with three widely cited datasets commonly used in the counterfactual explanation literature and one real-world dataset. **MNIST** (LeCun, 1998). In particular, we evaluate across several contrastive pairs of classes where counterfactual perturbations are intuitive to comprehend: digit 1 and 4, digit 1 and 7, digit 3 and 8, and digit 7 and 4. **German credit dataset** (ger) classifies people described by a set of attributes as good or bad credit risks. **Adult-Income** (Ronny & Barry, 1996) records whether a person makes over 50K a year based on census data. **Caltech-UCSD Birds (CUB)** (Wah et al., 2011). This is a real-world dataset for fine-grained bird classification. This dataset contains 200 bird species and we evaluate the counterfactuals across three contrastive pairs of bird species: Red Faced Cormorant and Crested Auklet, Myrtle Warbler and Olive sided Flycatcher, Horned Grebe and Eared Grebe. Bird species classification is a difficult problem that pushes the limits of the visual abilities for both humans and computers. Some pairs of bird species are nearly visually indistinguishable and intraclass variance is very high. This is arguably the most complex and high dimensional problem studied so far in the counterfactual explanation literature. To make this problem slightly more approachable, we work with feature representations extracted from the final convolutional layers of VGG-16 (Simonyan & Zisserman, 2015) pretrained on ImageNet. That is, we train a RAT-SPN as a generative classifier using the class-conditional feature maps.

**Baseline**: We consider three widely cited model-agnostic approaches in the literature that can be directly applied to our chosen datasets: Wachter et al., CEM (Dhurandhar et al., 2018) and FACE (Poyiadzi et al., 2020) [1].

**Implementation details**: To evaluate our approach with empirical evidence, we trained a RAT-SPN for each dataset. Each RAT-SPN is, as in figure 1, a mixture of class-conditional densities. This RAT-SPN is used for both classification and density estimation. For each dataset, the counterfactual approaches are evaluated on the same RAT-SPN classifier. We used cross-validation to select hyperparameters for RAT-SPNs and for all the counterfactual methods. See appendix for more details.

### 4.1 OUR COUNTERFACTUAL EXAMPLES ARE VISUALLY APPEALING.

Counterfactual examples can be presented to users as contrastive explanations. The examples that correspond to prior beliefs of the users can be better received by them due to confirmation bias (Nickerson, 1998). That means, the counterfactual examples should appear plausible and not deviate too far from the training samples. Figure 3 demonstrates some examples on MNIST for a variety of test cases across four methods. It is obvious to see that our approach consistently yields the most visually appealing examples as they are smooth, clean and look very plausible. Although being

---

[1] Since the real-world dataset is quite high-dimensional compared to those commonly used in the literature, we discarded some baselines with scalability issue after experimenting with it, e.g. DiCE (Mothilal et al., 2020).

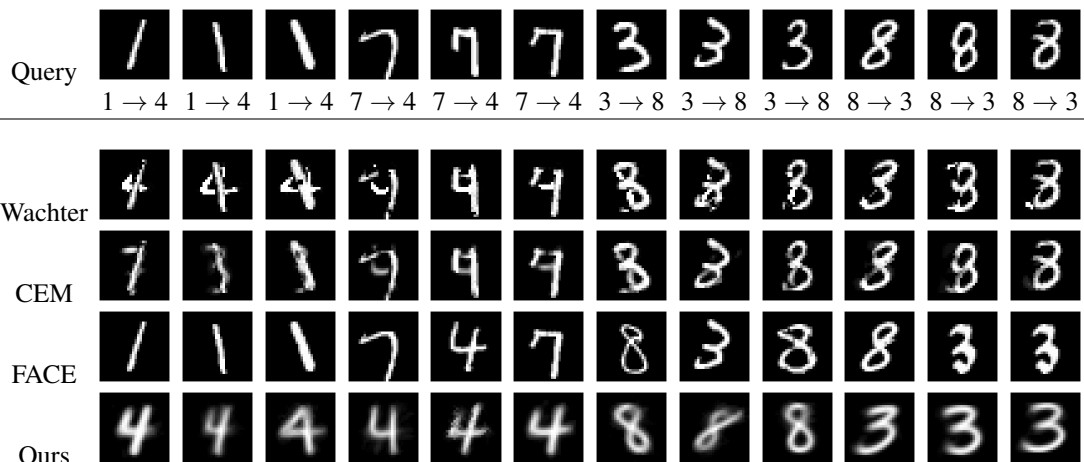

Figure 3: Counterfactual examples on MNIST across several classes and methods. The top row indicates the original class and the target class.

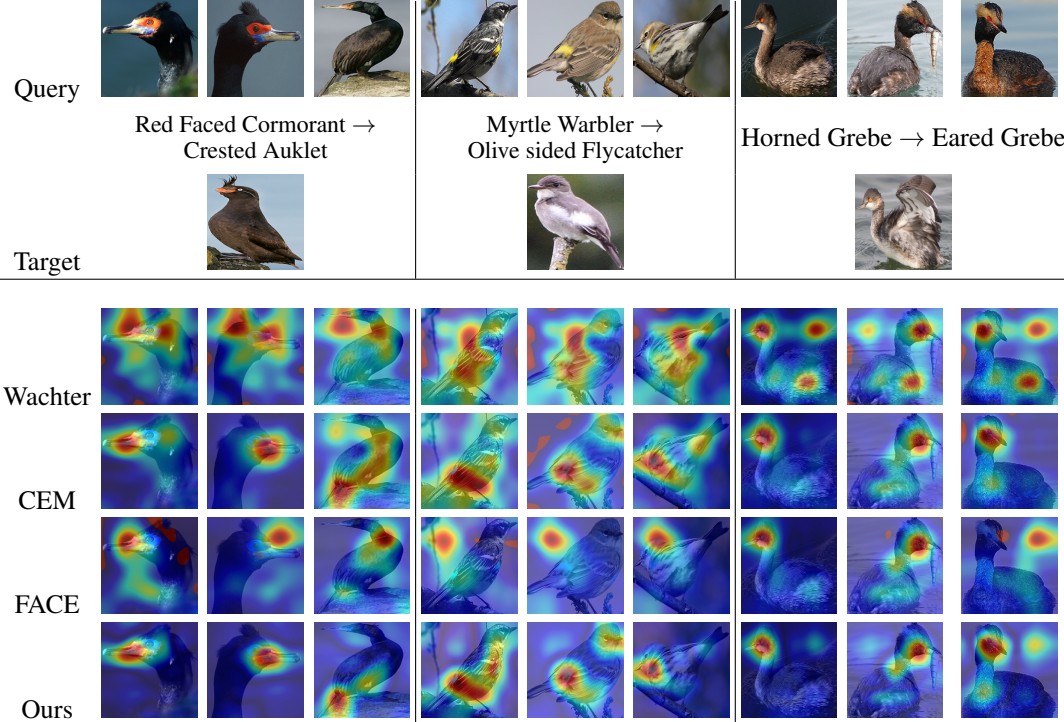

Figure 4: Counterfactual examples on Caltech-UCSD Birds (CUB) across several classes and methods. The top row indicates the original class and the target class.

smooth, our counterfactual examples still show a nice variation: For example, in the 7-th and 8-th column, the counterfactual 8 tilts to the left when the query image tilts slightly to the left, and likewise to the right. In comparison, Wachter et al. and CEM (Dhurandhar et al., 2018) both yield quite wiggly and noisy results. Although CEM includes a VAE reconstruction loss in its objective function as a proxy for constraining the examples to be in-distribution, explicit evaluation of density is intractable. FACE (Poyiadzi et al., 2020) also yields very plausible examples because it simply returns a training instance of the counterfactual class.

Another visual example can be seen on CUB. As previously said, we trained a RAT-SPN as a generative classifier using the class-conditional feature maps generated by VGG-16. In this case, the

RAT-SPN gives a density model on the feature maps and the counterfactual explanations are also computed in this extracted feature space. Since feature maps can not be trivially transformed back to raw features, we strive only for highlighting the salient counterfactual features instead of imputing them. Figure 4 demonstrates some test examples on this dataset. The heatmap overlaid on the example indicates the salient counterfactual features that should be perturbed to become the counterfactual class. An example from each counterfactual class is shown in the first row. Note that this is not the particular target we strive for but only an example for the readers to have an intuitive idea about the contrastive features. It can be seen that our approach consistently yields plausible heatmaps: For the Red Faced Cormoran the beak is highlighted when the image focuses on the head, and the tail is highlighted when the image zooms out. For the Myrtle Warbler the salient features are mostly on the yellow spot of the feather. For the Horned Grebe the salient features are mostly highlighted around the head. Among the baselines, CEM and FACE are implicitly density-aware and often show consistent behavior with our approach. However, FACE sometimes yields saliency on the background instead of on the bird. Wachter et al. performs the worst and yields quite random and unintuitive heatmaps. In conclusion, our approach showed very competitive and intuitively plausible heatmaps that highlight the contrastive features on this complex dataset.

From visual examples on both datasets we can see our approach yields appealing and intuitive results that are easy to comprehend.

## 4.2 OUR COUNTERFACTUAL EXAMPLES HAVE HIGH LIKELIHOOD.

In contrast to wiggly examples, smooth examples often appear more plausible and realistic, which are in turn often tied with high likelihood. This experiment is to offer a quantitative evaluation on likelihood.

It is widely agreed that out-of-distribution (OOD) counterfactual examples have very little use in communicating explanations with humans. Wachter et al. (2017) penalizes distance to the query instance, which indirectly also prevents the counterfactual examples to deviate too far from the distribution. More recent approaches use VAEs as a neural density estimator to penalize OOD examples (Dhurandhar et al., 2018; Ivanov et al., 2019; Klys et al., 2018; Joshi et al., 2019; Pawelczyk et al., 2020). FACE (Poyiadzi et al., 2020) takes a different approach and searches through the training samples directly for a counterfactual example instead of constructing it via an optimization process.

However, these approaches are not able to constrain density explicitly and directly because density evaluation is simply intractable. The direct consequence is that the proxy constraint does not consistently yield counterfactual examples in high-density region. To confirm this with empirical evidence, we take the advantage of RAT-SPNs on tractable inference to efficiently estimate and evaluate density.

Table 1 summarizes the average density evaluation on all the counterfactual examples across four datasets using four candidate approaches. On three of these datasets, our approach yields the best density among the baselines. FACE is a strong competitor in terms of likelihood because it always returns a training instance. But its limitation is obvious: It assumes we have access to the training samples, which is oftentimes not the case, e.g. due to privacy reasons.

To have some intuition on tabular data, see table 2 for some randomly selected examples on the German credit dataset. The columns from A10-1 to A15-3 are all one-hot encoded categorical attributes. For example, A10-1, A10-2 and A10-3 encode feature 10. Due to space constraint, only mutable features are shown. Each feature is perfectly negatively correlated with other features from the same categorical attribute due to the constraint imposed by one-hot encoding. Therefore we expect counterfactual perturbation to obey this feature correlation in order to stay close to the underlying distribution. That means, if one feature has positive perturbation, the rest features from the same attribute should have negative perturbation. That means, their perturbations should sum up to almost zero. Ideally each perturbation should be -1 or +1, but we work with continuous values in practice so this is often not the case. From table 2 one can see that our approach and FACE always respect this relation — the perturbations within each attribute always sum up to zero or nearly zero, showing an negative correlation. It is no surprise that FACE always perfectly reflects this constraint because its counterfactual examples directly come from the training set, which perfectly satisfy this constraint by construction. In contrast, Wachter et al. and CEM often violate this constraint: Take Wachter et al. as example, its perturbation on the first query yields 1, 1, -1 (sum up to 1) on

Table 1: Density evaluation in log scale averaged on the test set (the higher, the better). Best result indicated using ●, runner-ups ○.

|  | Wachter | CEM | FACE | Ours |
|---|---|---|---|---|
| MNIST | -738.51 | -735.13 | -733.73○ | -725.13● |
| Credit | -50.61 | -43.55○ | -43.83 | -41.67● |
| Adult | -114.20 | -96.99 | -96.26● | -96.42○ |
| CUB | -4593.85 | -4505.00 | -4501.99○ | -4501.23● |

Table 2: Counterfactual perturbations for German credit dataset with attributes encoded in on a one-hot fashion. Attribute 10: Other debtors / guarantors — A10-1 : none, A10-2 : co-applicant, A103 : guarantor, Attribute 14: Other installment plans — A14-1 : bank, A142 : stores, A14-3 : none, Attribute 15: Housing — A151 : rent, A15-2 : own, A15-3 : for free. A cell color "green" denotes a negative correlation that we expect from the counterfactual perturbation. A cell color "red" denotes a wrong correlation.

|  | Attribute 10 | | | Attribute 14 | | | Attribute 15 | | | $\hat{y}$ |
|---|---|---|---|---|---|---|---|---|---|---|
|  | A10-1 | A10-2 | A10-3 | A14-1 | A14-2 | A14-3 | A15-1 | A15-2 | A15-3 | |
| query | 0.00 | 0.00 | 1.00 | 0.00 | 0.00 | 1.00 | 0.00 | 1.00 | 0.00 | 0 |
| Wachter | 0.00 | 1.00 | -1.00 | 1.00 | 1.00 | -1.00 | 1.00 | -1.00 | 1.00 | 1 |
| CEM | 0.03 | 0.05 | -0.09 | 0.08 | 0.08 | -0.08 | 0.07 | -0.29 | 0.08 | 1 |
| FACE | 1.00 | 0.00 | -1.00 | 0.00 | 0.00 | 0.00 | 0.00 | 0.00 | 0.00 | 1 |
| Ours | 0.90 | 0.05 | -0.95 | 0.18 | 0.05 | -0.23 | 0.22 | -0.39 | 0.16 | 1 |
| query | 1.00 | 0.00 | 0.00 | 0.00 | 0.00 | 1.00 | 1.00 | 0.00 | 0.00 | 0 |
| Wachter | 0.00 | 1.00 | 0.00 | 1.00 | 1.00 | -1.00 | 0.00 | 0.00 | 1.00 | 1 |
| CEM | 0.00 | 0.00 | 0.00 | 0.00 | 0.00 | 0.00 | 0.00 | 0.00 | 0.00 | 1 |
| FACE | 0.00 | 0.00 | 0.00 | 0.00 | 0.00 | 0.00 | 0.00 | 0.00 | 0.00 | 1 |
| Ours | -0.10 | 0.05 | 0.05 | 0.18 | 0.05 | -0.23 | -0.78 | 0.61 | 0.17 | 1 |
| query | 1.00 | 0.00 | 0.00 | 0.00 | 0.00 | 1.00 | 0.00 | 1.00 | 0.00 | 0 |
| Wachter | 0.00 | 1.00 | 0.00 | 1.00 | 1.00 | -1.00 | 1.00 | -1.00 | 1.00 | 1 |
| CEM | 0.00 | 0.00 | 0.00 | 0.00 | 0.00 | 0.00 | 0.00 | -0.21 | 0.00 | 0 |
| FACE | 0.00 | 0.00 | 0.00 | 0.00 | 0.00 | 0.00 | 1.00 | -1.00 | 0.00 | 1 |
| Ours | -0.10 | 0.05 | 0.05 | 0.18 | 0.048 | -0.23 | 0.22 | -0.39 | 0.17 | 1 |

attribute 14. These perturbations are obviously not balanced. This intuitive example is related to their likelihood evaluation (see table 1): Those who respect the constraint better tend to yield higher likelihood because they are more realistic.

In conclusion, our counterfactual examples have dominant advantages on staying close to high-density region.

### 4.3 OUR APPROACH IS MUCH FASTER TO COMPUTE.

The baseline methods employ a methodology that's widely represented in the literature: Defining a complex objective function with additional constraints and use optimization techniques to iteratively find a solution. This is usually too slow to yield counterfactual examples on the fly. Especially when users want to interact with machine explanations, fast computation becomes more essential. As empirical evidence, table 3 shows that our method is an order-of-magnitude faster than the baseline approaches. This is no surprise due to the fact that our approach takes only two gradient steps while the baselines can easily take up to thousands of iterations.

### 4.4 OUR APPROACH IS EFFECTIVE.

The goal of counterfactual explanation is to find a contrastive example that changes the class prediction. An approach is effective if it has a higher success rate in perturbing the class prediction to the counterfactual class. This measurement is widely reported in the literature.

Table 3: Average computation time on the test set in seconds (the lower, the better). Best result shown in bold.

|  | Wachter | CEM | FACE | Ours |
|---|---|---|---|---|
| MNIST | 181 | 216 | 281 | **27** |
| Credit | 59 | 97 | 757 | **21** |
| Adult | 52 | 57 | 1722 | **10** |
| CUB | 65 | 468 | 63 | **16** |

Table 4: Success rate is measured by the ratio of counterfactual examples that yields the target prediction class (the higher, the better). Note that success for CEM is measured by any prediction perturbation for its own fairness, i.e. the perturbed prediction does not need to be the specified target class.

|  | Wachter | CEM | FACE | Ours |
|---|---|---|---|---|
| MNIST | 1.00● | 0.90○ | 0.54 | 0.71 |
| Credit | 1.00● | 0.87 | 0.92 | 1.00● |
| Adult | 1.00● | 0.93 | 0.50 | 0.99○ |
| CUB | 1.00● | 0.73 | 0.90 | 0.98○ |

Except for CEM, all the success rate are measured by the ratio between examples with the counterfactual prediction and all the test examples. Since CEM encourages a contrastive example belonging to any other class than the original class, measuring its success only by a specific class prediction is not fair for CEM. Therefore we measure its success rate by the ratio between examples with perturbed class prediction and all the test examples. This metric is ranged between 0 and 1 where 1 is the best and 0 is the worst.

Table 4 gives a summary of success rate. Wachter et al. is very effective at perturbing the class prediction, with a success rate of 1.0 across various datasets. CEM and FACE are less effective. Our approach has slightly less success rate than Wachter et al. but still very effective in general with a very reasonable success rate. This result is not surprising: Wachter et al. has the fewest constraint in its objective function, while CEM and our approach face a trade-off between prediction success and density constraint.

In conclusion, although being so fast to compute, our approach does not sacrifice the effectiveness of perturbing the class prediction.

## 5 CONCLUSIONS AND FUTURE WORK

In this work we presented a novel way of generating counterfactual examples using tractable probabilistic inference. The core idea is to view counterfactual example generation as a two-step process: First perturb the class prediction irregardless of the underlying distribution by maximizing the conditional likelihood of the counterfactual class. As this would probably result in an unrealistic counterfactual example that is far away from the underlying distribution, we then maximize its likelihood in the second step by taking a step towards the direction of the gradient of the likelihood function. This is possible because density evaluation is tractable for the probabilistic models we use. Our approach is not only very effective for generating counterfactual examples, but also very fast to compute. In addition, the counterfactual examples have high likelihood.

One interesting direction to investigate in future work is generating counterfactual examples interactively based on human feedback. Another direction is to enforce more complicated or specific constraints on the counterfactual examples. As a highly tractable generative model, SPNs allow for a wide variety of tractable probabilistic inferences, which could be used to formulate more complex constraints. It is also interesting to incorporate human supervision on counterfactual explanations to improve the underlying classification model. All of these research directions would not be approachable with the existing approaches.

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

## 6 APPENDIX

### 6.1 IMPLEMENTATION DETAILS

RAT-SPNs (Peharz et al., 2020) is a simple approach to learn an SPN in a deep learning fashion: constructing a random SPN structure and learn the parameters via gradient-based techniques. To construct the random SPN structure, Peharz et al. (2020) use the notion of a region graph as an abstract representation of the network structure. Given a set of random variables (RVs) $\mathbf{X}$, a region $R$ is defined as any non-empty subset of $\mathbf{X}$. Given any region $R$, a $K$-partition $P$ of $R$ is a collection of $K$ non-overlapping sub-regions $R_1, \ldots, R_K$, whose union is again $R$, i.e. $P = \{R_1, ..., R_K\}, \forall k : R_k \neq \emptyset, \forall k \neq l : R_k \cap R_l = \emptyset, \cup_k R_k = R$. In practice, only 2-partitions are considered. To construct random regions graphs, we randomly divide the root region into two sub-regions of equal size (possibly breaking ties) and proceed recursively until depth $D$, resulting in an SPN of depth $2D$. This recursive splitting mechanism is repeated $R$ times. Therefore, the size of RAT-SPNs are controlled by the following structural parameters: *split-depth $D$*, *number of split repetitions $R$*, *number of sum nodes in regions $S$*, *number of input distributions per leaf region $I$*.

In our experiments, we trained a RAT-SPN for each dataset and we set $D$ to 1. This RAT-SPN has dual use: It is both a density estimator $\mathcal{S}(\mathbf{X})$ and a classifier $\mathcal{S}(Y|\mathbf{X})$. The rest tuning-parameters are determined via cross-validation. We cross-validated $R \in \{19, 29, 40\}$, $S \in \{2, 10\}$, $I \in \{20, 25, 33\}$.

We divided each dataset into 70%-30% train and test sets where train set was used to train the SPNs and optimize tuning-parameters via cross-validation. Test set was used to test counterfactual examples. The baseline Wachter et al. (2017) and CEM both used gradient-based optimization combined with Adam optimizer. Early stopping is implemented to stop early when all the query examples are perturbed to the counterfactual class. For FACE, we used only the first 1k training examples for searching in order to maintain a reasonable computation time.

**MNIST Dataset**: We scaled all features to the range between 0 and 1. Since only digit 1, 3, 4, 7 and 8 are used in the experiment, we used only those images to train an SPN, i.e. the SPN has 5 classes. Test accuracy of this SPN on the class prediction task is 98%. The reported results are based on the following hyperparameters: $R = 19$, $S = 10$, $I = 20$. We also cross-validated hyperparameters for Adam optimizer used in Wachter et al. (2017) and CEM: learning rate $\in \{0.5, 0.05\}$. Final results use learning rate=0.5, epochs=5000. For CEM, we set $\beta = 1$ and cross-validated $c \in \{10, 100\}$ and $\gamma \in \{0.1, 1\}$. The reported results use $c = 100$ and $\gamma = 0.1$. For FACE, we used mode = 'KNN' where $k = 5$. For our approach, we set $\epsilon_2 = 1$ and cross-validated $\epsilon_1 \in \{1, 10\}$. The reported results use $\epsilon_1 = 10$.

**CUB Dataset**: We scaled all images to $224 \times 224 \times 3$ so it can be used for VGG-16. We report results with the following hyperparameters. The last convolutional layer of VGG-16 has dimension $7 \times 7 \times 512$ and we use the first 100 feature maps for a speedup. That is, the SPN takes input of $7 \times 7 \times 100$ features. We did the same cross-validation as for MNIST and report the results for the following hyperparameters: For RAT-SPN, $R = 29$, $S = 10$, $I = 25$. For Wachter et al., learning rate = 0.05, epochs = 1000. For CEM, learning rate = 0.5, epochs = 7000. For FACE, we used mode = 'KNN' where $k = 5$. For our approach, $\epsilon_1 = 10$ and $\epsilon_2 = 1$.

**Adult Dataset**: We standardized features by removing the mean and scaling to unit variance and transform categorical features by using one-hot-encoding. We report results with the following

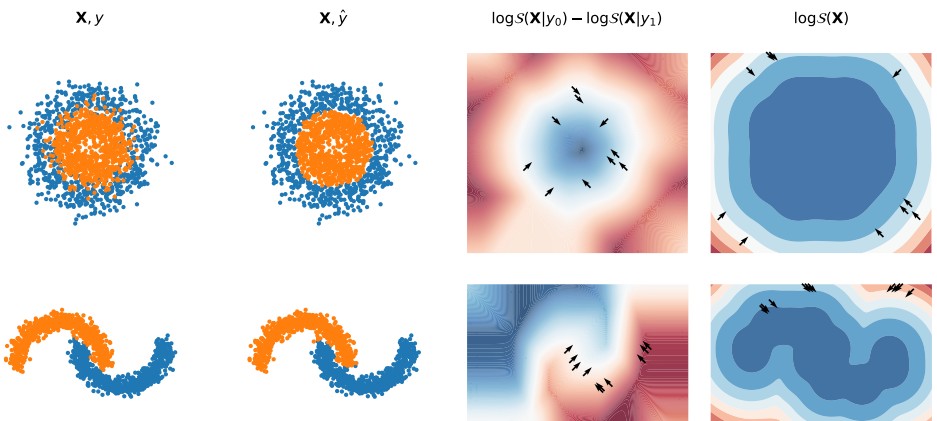

Figure 5: Individual illustration of the two gradients used in our approach on two commonly used 2D datasets. From left to right, the figures are: the training data for classification task, SPN's prediction on this set, the contour lines of SPN's decision boundary, i.e. $\log \mathcal{S}(\mathbf{X}|y_0) - \log \mathcal{S}(\mathbf{X}|y_1)$, and its gradient, the contour lines of SPN's density on $\mathbf{X}$. The orange data points are class $y_0$ and the blue data points are $y_1$. Note that each figure on a row are plotted under the same scale, and the gradient in separate figures do not have one-to-one correspondence.

hyperparameters: For RAT-SPN, $R = 19$, $S = 10$, $I = 20$. This RAT-SPN gives a test accuracy of 74%. For Wachter et al., learning rate = 0.05, epochs = 1000. For CEM, learning rate = 0.5, epochs = 7000. For FACE, we used mode = 'KNN' where $k = 5$. For our approach, $\epsilon_1 = 10$ and $\epsilon_2 = 1$.

**German Credit Dataset**: We transformed features by scaling each feature to the range between 0 and 1 and transform categorical features by using one-hot-encoding. We report results with the following hyperparameters. We selected the following features to use: 'checking status', 'history', 'purpose', 'savings', 'employ', 'status', 'others', 'property', 'other plans', 'housing', 'foreign', 'age', 'amount', 'duration'. For RAT-SPN, $R = 40$, $S = 10$, $I = 33$. This RAT-SPN gives a test accuracy of 69%. For Wachter et al., learning rate = 0.05, epochs = 1000. For CEM, learning rate = 0.5, epochs = 1000. For FACE, we used mode = 'KNN' where $k = 5$. For our approach, $\epsilon_1 = 10$ and $\epsilon_2 = 1$.

## 6.2 2D INTUITION

In figure 5, we plot the two gradients used in our approach on two commonly used 2D datasets to given an intuition. We trained a RAT-SPN on each dataset, one can see in the third column that the decision boundary of RAT-SPN is given by $\log \mathcal{S}(\mathbf{X}|y_0) - \log \mathcal{S}(\mathbf{X}|y_1)$. Therefore, taking a gradient of $\log \mathcal{S}(\mathbf{X}|y_0) - \log \mathcal{S}(\mathbf{X}|y_1)$ w.r.t the input induces a direction for crossing the decision boundary under first-order approximation around the query example. The gradient vectors are visualized by arrows, and the gradient in the third column are computed on randomly chosen test examples of class $y_1$ (from the blue cluster in the training set). In case the dataset is not dense in the input space, like the second dataset, the first gradient step may very likely extrapolate to a low-density region. Fortunately, the second gradient step of $\mathcal{S}(\mathbf{X})$ serves as a first-order approximation of the local density and takes an example in low-density region to higher-density region. See the last column, the gradient vectors are computed on randomly chosen *out-of-distribution* examples.

