# OpenReview forum: "Gradient-based Counterfactual Explanations using Tractable Probabilistic Models"
_ICLR.cc/2022/Conference — ICLR 2022 Submitted_

### Official Review · Reviewer_Nam5 · 2021-10-31

**Correctness:** 2
**Technical Novelty And Significance:** 2
**Empirical Novelty And Significance:** 3
**Recommendation:** 3
**Confidence:** 2

**Main Review:**

Overall, the idea of using SPNs for trustworthy machine learning models is legitimate, and the algorithm proposed by the authors for inferring counterfactual explanations from SPNs is both theoretically simple and empirically effective.

Unfortunately, it is quite difficult to accept the paper in its current state, due to clarity issues in both the theoretical part (Section 3) and the experimental part (Section 4).

In the theoretical part of this study, we essentially have a brief presentation of SPNs, together with a two-step gradient-based algorithm for inferring counterfactual explanations. However, what problem is solving this algorithm? Notably, are the counterfactual explanations returned by this algorithm satisfying some desirable properties? Furthermore, what is the convergence rate of this algorithm?

These questions call for a more formal presentation of the technical part of this study. As we know, counterfactual explanations are not just perturbations of the input data instance that change the output class. The literature on XAI has identified several properties which should be satisfied by counterfactual explanations. Among them are validity, actionability, sparsity, causality, etc. (Verma, Dickerson, and Hines, 2020). Each subset of these properties is associated with a well-defined, formal optimization problem. Thus, in order to clarify the behavior of the two-step gradient algorithm presented on Pages 4-5, the underlying optimization task should be appropriately formulated. Once we have a clear problem formulation with the desired properties of counterfactual explanations, the behavior of the two-step gradient algorithm should be analyzed in order to have some formal guarantees on its convergence rate.
My guess is that the judicious choice of SPNs as machine learning models allows for tractable forms of counterfactual explanations satisfying at least the validity property (i.e. L2-distance minimization), and the actionability & causality properties (due to the fact that the returned counterfactual explanations have high likelihood).

In the experimental part of this study, we are not comparing the same “objects of study”. On the one hand, we have some model-agnostic approaches to counterfactual explanations (Wachter et. al., CEM, FACE). So, these model-agnostic approaches can be applied to many candidate ML models, requiring only a few assumptions about the model (e.g. Wachter et. al. only require access to gradients). On the other hand, we have a model-based approach to counterfactual explanations (this study), which focuses on SPN models. Since the authors are comparing different counterfactual explanations methods on different classes of machine learning models, the prediction performance of the models should also be reported. For example, using the Adult-Income dataset, the classification performance of RAT-SPN is approximately 74% (as reported in the Appendix); but what is the classification performance of the model upon which the Wachter et. al. explanation method has been applied? My point here is related to the evidence that predictive performances should be comparable (in order to compare different counterfactual inference approaches), and a table highlighting these performances would be helpful. For reproducibility reasons, it would be nice to provide a link to some repository, with all source codes and benchmarks used in the experiments.

-- Reference used in this review:

Sahil Verma, John P. Dickerson, Keegan Hines: Counterfactual Explanations for Machine Learning: A Review. CoRR abs/2010.10596 (2020)


**Summary Of The Paper:**

Explaining in a human-understandable way the predictions made by machine learning models is an important research topic for the development of trustworthy systems. In this topic, the present paper focuses on counterfactual explanations, a specific class of explanations that provides a link between what could have happened if the input to a model had been changed.

The key idea of this paper is to exploit Sum-Product Networks (SPNs), a class of machine learning models capable of providing counterfactual explanations endowed with appealing computational properties. Based on those models, the authors present a conceptually simple two-stage gradient-based algorithm for computing counterfactual explanations. This approach is empirically compared with baseline techniques on several benchmarks.


**Summary Of The Review:**

As indicated above, the idea of using SPNs for generating counterfactual explanations in an effective way is relevant, and experimental results look promising. So, I encourage the authors in pursuing this avenue of research. However, for the moment, the paper has clarity issues, related to both the technical part and the experimental part. I think that the technical issues are most important.

---

> ### Author Response · Authors · 2021-11-18
> **Response to Reviewer Nam5**
>
> Thank you for your comments! We are glad to address your questions.
>
> 1. Our approach does not depend on an iterative optimisation process, this is one of the major feature and advantage of our approach. Like we mentioned in the manuscript, most of the counterfactual methods reframe the problem of generating counterfactuals as a black-box optimisation task, this deep-learning like approach essentially use another black-box approach to explain a black-box model, which we think does not help much in terms of interpretability and increasing trust. If this black-box explanations yield a wrong result, it would be difficult to notice and almost impossible to debug. In contrast, our approach takes a much more transparent and interpretable way to generate counterfactuals: each of the two steps have a clear semantics and the intermediate step can also be visualised and easily understood by human. We do not see any particular reason why a new approach that takes a novel perspective of reframing the task is a problem. Besides, although having a formal formulation of optimisation problem, most of the existing approaches do not have a theoretical analysis of their convergence rate or theoretical guarantee. Quite often, they deal with highly non-linear objective functions, which are very tricky to optimise and often arrive at unsatisfactory results. We added a new figure (figure 5) in the manuscript’s appendix to give further intuition about the two gradient steps of our approach on 2D datasets.
> 2. Validity refers to how effective is a method at inducing the desired counterfactual class, not about the distance. This is evaluated in section 4.4 by the success rate, which measures how many counterfactual examples arrive at the desired class. We aligned our empirical evaluation with the mainstream literature by reporting the metrics widely agreed and benchmarked in the literature. Of course, we agree additional properties like actionability and causality are probably important for counterfactuals, and they should generally be considered, but there is not yet a widely used benchmark, or widely agreed proxy to quantitively evaluate them. In fact, likelihood we reported in the evaluation section is also used as proxy for actionability in some literature [1], because it is probably more realistic to act on more likely examples than less likely ones. The density distribution can imply some complex interaction between features, which in turn constrain the counterfactuals to be realistic. This can be seen in our experimental example in table 2. Take another example in loan application [2], a counterfactual that tells a decision subject to increase his educational degree while keeping the age constant or even decreased may lead the example to low-density region (because the major population with higher educational degree in the training set also are older), thus not actionable. In contrast, a high likelihood counterfactual would require a increase of both educational degree and age, hence more actionable. In addition, these "more advanced" properties may sometimes be in conflict with each other, and it is debatable which property actually yields "better" counterfactuals. For example, sparsity (often evaluated via vector norm) is not always aligned with "better" counterfactuals. As Poyiadzi et al. [1] argued, a solution with small norm leading to “unachievable goals” is less optimal than a solution with bigger norm leading to feasible goals. They further used likelihood as proxy for the feasibility/actionability property. The likelihood can also serve as a proxy for prior beliefs. As Nickerson [3] argued, humans exhibit confirmation bias, meaning that we tend to ignore information that is inconsistent with our prior. All this discussion about the properties for counterfactuals and their proxy metrics is actually orthogonal to our work and deserves dedicated research, and here our goal is just to stress that we evaluated under the commonly agreed metrics and benchmarks with relatively few controversy.
> 3. There is a misunderstanding here about the experimental study. We do compare the same object of study. As we mentioned in the manuscript, on each domain, all the counterfactual methods are employed to exactly the same classifier, which is a RAT-SPN. So the evaluation is absolutely fair to compare. We apologise for the confusion, and we made some additional statement in the manuscript. Besides, we are glad to publish code upon acceptance for reproducibility.
>
>
> [1] Rafael Poyiadzi, Kacper Sokol, Raul Santos-Rodriguez, Tijl De Bie, and Peter Flach. Face: Feasible and actionable counterfactual explanations. In Proceedings of the AAAI/ACM Conference on AI, Ethics, and Society.
>
> [2] Ramaravind K Mothilal, Amit Sharma, and Chenhao Tan.  Explaining machine learning classifiers through diverse counterfactual explanations.
>
> [3] Raymond S Nickerson. Confirmation bias: A ubiquitous phenomenon in many guises. Review of general psychology.

---

### Official Review · Reviewer_eBHW · 2021-11-02

**Correctness:** 3
**Technical Novelty And Significance:** 3
**Empirical Novelty And Significance:** 3
**Recommendation:** 5
**Confidence:** 5

**Main Review:**

This paper introduces a counterfactual explanation method that relies on a two-step perturbation. The first perturbation nudges the original data instance towards the counterfactual class. The second perturbation uses sum-product networks to guide the counterfactual instance towards higher density regions and thus produce more realistic examples.

The approach has a number of attractive aspects including faster computation time than other deep learning counterfactual methods, which would enable it to be used interactively. In addition, the proposed method produces more realistic counterfactual instances relative to the other methods mentioned in the evaluation.

I agree with the authors that the optimization problem for deep learning counterfactual methods is tricky and they can produce unrealistic counterfactuals. The use of a sum product network is a nice idea and a promising direction.

There are, however, two major issues with this paper. First, the authors are unaware of current counterfactual methods that are closely related (see references at the end of this review). This paper needs a direct comparison to these techniques or a discussion of how the proposed approach improves on these state of the art counterfactual approaches. In light of these uncited papers, several of the claims regarding the novelty of the paper are not true. For instance, in the introduction, the authors state that they "...experiment with a complex and high-dimensional real-world dataset, which was not dealt with in the literature." but Goyal et al. (2019) used the CUB dataset in their evaluation.

Second, the evaluation section could be stronger. The authors show counterfactual images produced by their method and by other competitors. However, while some algorithms produce very obviously low quality counterfactuals, the images produced by the method in this paper sometimes picks up on regions in the background of the bird (like in FACE). In addition, I found the MNIST counterfactuals produced by the authors' method to be noticeably blurry; is there a reason for this blur? As a result, the proposed method is not a clear improvement over other methods on images and it is also an open question of how it would compare against the recent counterfactual methods that were not cited.

The authors also show that their method produces higher likelihood images and that the method is fast in terms of compute time. These qualities of their method are good, but the key question that is not answered is whether the counterfactual explanations are good quality explanations to a human and whether they are better explanations than other state of the art counterfactual methods. To answer this question, the authors need to perform a user study to evaluate how humans would respond to the explanations. Such a user study is non-trivial to design and requires careful thought as to how the explanations can be used by a human to perform a task.




References
----------------
M. Olson, R. Khanna, L. Neal, F. Li  and W-K. Wong. (2021). Counterfactual State Explanations for Reinforcement Learning Agents via Generative Deep Learning. Artificial Intelligence, 295, doi: 10.1016/j.artint.2021.103455

Yash Goyal, Ziyan Wu, Jan Ernst, Dhruv Batra, Devi Parikh and Stefan Lee. Counterfactual visual explanations. International Conference on Machine Learning, ICML (2019).

Chun-Hao Chang, Elliot Creager, Anna Goldenberg and David Duvenaud. Explaining image classifiers by counterfactual generation
International Conference on Learning Representations (2019).


**Summary Of The Paper:**

The paper present a new method for counterfactual explanations that relies on a two-step perturbation of the original data instance. The first step moves the original image towards the counterfactual class while the second step uses a sum product network to guide the counterfactual instance towards a higher density region.


**Summary Of The Review:**

The paper presents a novel counterfactual explanation method based on sum product networks. While the method has some attractive properties, the paper needs to compare against recent counterfactual explanation methods and the evaluation section needs to be more compelling.

---

> ### Author Response · Authors · 2021-11-18
> **Response to Reviewer eBHW (part 2)**
>
> We also acknowledge your comments about CUB dataset, we rephrased in the updated manuscript. About the quality of our counterfactual examples, the only one counterfactual image from our approach that picked up on the background picked up much more saliency intensively on the birds head, according to the heatmap. Compared to this very little noise in the heatmap, all the baselines have also more or less noisy heatmaps on some examples. So this is not a good reason to downplay our approach. About MNIST, our counterfactuals are more smooth, this is very likely related to the feature of SPNs. Each leaf node is a univariate Gaussian distribution, which are hierarchically composed together to yield the joint distribution. Taking a gradient step towards the density ascent is in fact taking a step towards the centre of a local Gaussian blob (where the density is highest in the neighbourhood). And when the parameters are estimated with MLE, the mean of a gaussian takes the mean of the training data, which explains the smoothing effect. In general, we do not consider the smoothing effect a negative thing, and we argue that our counterfactuals are more much more plausible than many ‘sharp’ but wiggly baselines. In fact, our counterfactuals do also have higher likelihood according to quantitative evaluation. We agree that having user study would be a nice supplement, and one should definitely do it if time and resource allows, but this is not a indispensable or standard component in the counterfactual literature, see the most cited examples: Wachter et al.[1], FACE[2], DiCE[3], Ustun et al.[4]. Our experiments report on commonly used metrics in the literature, and these metrics are commonly agreed proxy for evaluating the quality of counterfactuals. As for if these metrics are actually good proxy for evaluation indeed deserves dedicated research attention and large scale user study, but this is rather orthogonal to our work.
>
>
> [1] Sandra Wachter, Brent Mittelstadt, and Chris Russell. Counterfactual explanations without opening the black box: Automated decisions and the gdpr. Harv. JL & Tech.
>
> [2] Rafael Poyiadzi, Kacper Sokol, Raul Santos-Rodriguez, Tijl De Bie, and Peter Flach. Face: Feasible and actionable counterfactual explanations. In Proceedings of the AAAI/ACM Conference on AI, Ethics, and Society.
>
> [3] Ramaravind K Mothilal, Amit Sharma, and Chenhao Tan. Explaining machine learning classifiers through diverse counterfactual explanations. In Proceedings of the 2020 Conference on Fairness, Accountability, and Transparency.
>
> [4] Berk Ustun, Alexander Spangher, and Yang Liu. Actionable recourse in linear classification. In Proceedings of the Conference on Fairness, Accountability, and Transparency.

---

> > ### Comment · Reviewer_eBHW · 2021-11-21
> > **Comments on Reponse to Reviewer eBHW**
> >
> > I thank the authors for their detailed response, which clarified some of my questions.
> >
> > Although I agree that SPNs are a promising approach to generating counterfactuals, I feel that the paper is not quite ready and needs a substantial revision of the evaluation section. I will maintain by review rating. Comments are below:
> >
> > 1) I agree with the authors that a nice benefit of their approach is that it is not restricted to image data but is much more general. However, roughly half of the evaluation section is devoted to image datasets (MNIST, CUB). Despite the authors arguing that the papers I cited are not applicable because they are specific to vision tasks, the authors use vision tasks to show the superiority of their approach in the experiments. I recommend that the authors include more results from non-image datasets to highlight the generality of their approach.
> >
> > 2) If the authors want to include image datasets in their evaluation, I feel they should compare against a state-of-the-art technique like Goyal et al. (2019). I disagree that it is not a sensible baseline -- the distractor image is straightforward to set up as a head-to-head comparison with the author's SPN approach. In their response, the authors point out issues with the counterfactual approach of Goyal et al. (2019). It would be very compelling to include concrete examples where the SPN-based method produces obviously better counterfactuals than the approach by Goyal et al. (2019). Also, note that the approach by Olson et al. (2020) is not restricted to RL as it can be applied to supervised learning for image data.
> >
> > 3) I liked Section 4.2, which is one of the more compelling results. On the whole, however, I feel that many of the commonly used metrics (e.g. success rate, validity) capture isolated aspects of counterfactuals but don't actually tell the whole story regarding the usefulness of a counterfactual. While a user study is not always required for these types of XAI papers, it does directly measure the effectiveness of the counterfactuals (provided the study is designed correctly) and puts a lot of concerns to rest. Note that the Goyal et. al. (2019) and Olson et al. (2020) papers include the results of user studies.
> >
> > 4) The section could also use a discussion on the limitations of the approach. The blurry images on MNIST indicate that the Gaussian approximation at the leaf may be too coarse of an approximation for some datasets. It would be informative to include a discussion (or experiment) of how the counterfactual quality is affected by how well the SPN approximates the true density of the data.

---

> > > ### Author Response · Authors · 2021-11-22
> > > **Response to Reviewer eBHW**
> > >
> > > Thank you for your comments! We hope to clarify some misunderstanding here.
> > >
> > > 1. The claim that "roughly half of the evaluation section is devoted to image datasets" is not true at all, actually only one section (4.1) out of four focused on the visual effect, and all the rest 3 sections are evaluated generally across all the datsets, including two tabular datasets, and intuitive understanding is also given for one tabular data.
> > > 2. We need to stress that Goyal et al. are used under very different assumption and context than our approach, therefore one can not compare the same “objects of study”. It is not about feasibility, but a fair comparison. In general, Goyal et al. use more prior knowledge and assumptions about the domain to solve the task. I think one thing is already obvious with the current available results, our counterfactuals have more flexible shape and size, while Goyal et al. assume a fixed shape and size for their counterfactuals. The distractor images used by Goyal et al. give also additional information to the counterfactual method, which we do not use. The fact that we use the same dataset as Goyal et al. does not imply they are the sensible baseline for us. Actually, many domain-agnostic counterfactual literature after Goyal et al. also do not consider this as SOTA baseline on the MNIST dataset. So we think it is not a good reason to downplay our approach.
> > > 3. As we said, whether the commonly used metrics are good proxy for evaluation in terms of human experience, requires dedicated research attention and is rather orthogonal to our work. The work that include user study, like Goyal et. al. (2019) and Olson et al. (2020), also do not report on the commonly used metrics. So user study is rather an alternative than a complement to the commonly used metrics. And arguing for one alternative against another popular methodology requires solid argument and study, otherwise it is rather a personal preference, therefore this is not a good reason to downplay our evaluation as well.
> > > 4. The gaussian leaves can be potentially refined by using smaller variance. But the current setting did not impair the experiment results so far, therefore the refinement can be left for future work. We added an additional toy experiment in the appendix (figure 5) to illustrate how our approach works on 2D datasets.

---

> ### Author Response · Authors · 2021-11-18
> **Response to Reviewer eBHW (Part 1)**
>
> Thank you for pointing out the references! We are glad to address your questions.
>
> We acknowledge the three references and added citation to them in the updated manuscript. However, they are not sensible baselines for our approach. In the following we explain the reasons one by one.
>
> 1. First of all, Olson et al. is dedicated for RL problem and our method focuses on supervised learning tasks, so it is not directly comparable. Second of all, this work neither compared with any widely cited baseline methods in the literature, nor used any standard benchmark datasets for counterfactual explanations, so it is not yet clear that this work provides state-of-the-art performance.
>
> 2. For Goyal et al., this approach focuses on vision tasks, which implicitly incorporates many assumptions specific for vision tasks: first, both a query image and a distractor image of the counterfactual class are assumed for Goyal et al., but we are based on only a query and not restricted to vision domain. (Note that target images in figure 4 is given only for illustrative purpose, these particular images are not used in any way for counterfactual generation.) Second, the counterfactual of Goyal et al. is one connected area of pixels of fixed shape and size (due to the fixed size of the feature representation after convolutional layer), which is a very strong assumption. Note that their reported images of CUB (figure 8) have the bird all relatively at the same position and same scale. In case of a image zoomed in to the bird’s head, like in our figure 4, this approach will fail because their counterfactual can not adjust its size to cover a larger area of features. Hence, the choice of the distractor image is most likely also very crucial for the final counterfactual explanation. Thirdly, this approach assumes a convolutional layers as feature extractor, while our approach does not have this restriction. In conclusion, this approach is proposed under a much more restrictive context than our approach, in turn not applicable for our experimental setting.
>
> 3. For Chang et al., it is closely related to Goyal et al., both of them are designed for vision tasks. And they are not common baselines used in the literature for evaluating domain-agnostic methods. Since our approach is domain-agnostic, we considered the most cited and recent domain-agnostic approaches in the literature as standard baselines, so the experiments do provide a sensible evaluation. In addition, our approach solves a more fine-grained task than Chang et al. in that we generate examples of a desired counterfactual class y’ in contrast to y, whereas Chang et al. only highlight the salient features for class y.

---

### Official Review · Reviewer_LeuZ · 2021-11-02

**Correctness:** 4
**Technical Novelty And Significance:** 3
**Empirical Novelty And Significance:** 3
**Recommendation:** 8
**Confidence:** 4

**Main Review:**

The paper seems to clearly improve the SOTA in the field of counterfactual examples identification. This is definitely true in terms of execution times, and the quality of the generated examples seems to be high according to various performance descriptors. The topic has been subject of intense research in the last four years and I believe the paper can be safely accepted at ICLR.

I have found the description of the method in sect. 3 clear and easy to grasp, but adding a detailed description of some "deep learning" method could be useful to clearly understand the computational advantages of the current approach.

As execution times are the most notably advantages, more details should be provided about that part in the experimental section and the different results of the other methods on the different datasets should be commented.

I am not sure I understand the notion of "success". In my current understanding, if the value should be always one for any method, otherwise we just don't get what we want for the explanation. More details should be added there.

I also miss some discussion about the choice between methods such that the contrastive sample is lying in the training set (as in Face) and "virtual" ones. I would say that such choice might depend on the particular problem under consideration. I see this is not a specific issue of the current method, but a discussion on this point would help.

I should also say that, unlike the MNIST, the CUB experiment was less clear to me.


**Summary Of The Paper:**

This is a XAI paper presenting a method to find counterfactual examples (i.e., contrastive example leading to a different prediction from the one currently obtained for a given example). The approach is based on tractable generative probabilistic circuits. This makes the procedure considerably faster than the existing SOTA approach. This seems to be very valuable and extensive validations show good performances even in terms of interpretability, smoothness and effectiveness.

**Summary Of The Review:**

Good paper on a novel and important topic: a XAI algorithm to return counterfactual examples achieved by a generative approach based on probabilistic circuits. Results are in line with the SOTA and considerably faster in terms of execution times.

---

> ### Author Response · Authors · 2021-11-18
> **Response to Reviewer LeuZ**
>
> Thank you for your comments! We are glad to address your questions.
>
> Indeed, we have a huge advantage in terms of execution time. The baseline approach Wachter et al. and FACE are both based on iterative optimisation, which took many epochs to find valid counterfactuals, and each epoch requires a full back propagation, which explains the long execution time. FACE searches through the training set to find counterfactual examples, which usually also takes many iterations. Besides, its execution time vary a lot due to the search algorithm depending on the underlying domain and the test samples.
>
> A counterfactual is counted as successful if it induces the desired counterfactual class y’. And the success rate is the ratio of how many examples have arrived at class y’ out of all the query examples. This measures how effective the approach is at perturbing the prediction, which is the main goal of counterfactual examples. Therefore we report this metrics in section 4.4.
>
> Choice between counterfactual methods: FACE searches through the training set for a counterfactual, and thus assumes access to the training set. In addition, FACE have difficulty searching beyond the class manifold thats adjacent to the given query example. Therefore, FACE may not be used when training data is not accessible due to e.g. privacy reasons, and FACE may not consistently work when there are more than two class categories. CEM uses a VAE to approximate the input distribution, therefore it can only be used if a reasonably well trained VAE on the domain is available. Our approach assumes a RAT-SPN as density estimator, therefore it can not be used if RAT-SPN is not given. Wachter et al. has the least restriction, therefore it can almost always be used as long as computation budget allows. All these approaches are generally agnostic to domains, therefore they can be applied to vision tasks, tabular data, or textual data.
>
>
> For the CUB experiment, we learn a SPN on top of the features extracted by convolutional layers, then all the counterfactual are computed on this latent feature space.  Because the underlying SPNs are not modelled over the full distribution on the raw input space, therefore reconstructing counterfactual features are not feasible for any approach. In other words, we can not infill the raw features by its latent features, hence we map only the saliency from the latent feature space to the raw input space. And we use the heatmap overlaid on each example to highlight the counterfactual features, instead of generating concrete counterfactual examples (as is done for the MNSIT experiment). The counterfactual features highlighted by heatmaps indicate the features that are contrastive to a desired counterfactual class.

---

### Official Review · Reviewer_Hebe · 2021-11-03

**Correctness:** 3
**Technical Novelty And Significance:** 2
**Empirical Novelty And Significance:** 2
**Recommendation:** 3
**Confidence:** 3

**Main Review:**

There are a few questions that need further clarifications:

1. In page 5, line 3: the authors state: "u is very likely to deviate from the underlying data manifold". I wonder whether this claim is true. The way u is generated already captures the data likelihood via the conditional likelihood S(x|y') and S(x|y'). In my opinion, if u deviates, that basically means either (i) epsilon_1 is too big, or (ii) S is a wrong model. What else can be the root cause of the deviation of u from the data manifold?

2. To generate u, why do we need to minus S(x|y)? Isn't the goal just about maximizing S(x|y'), and there is no need to minimize S(x|y)?

3. It is not clear how only 2 gradient ascent steps (generate u and then x') would suffice. The paper seems to omits the constraint that x and x' should be close (in some distance). Should we minimize epsilon_1, while trying to meet some requirements (effectiveness, etc.)?

4. How should epsilon_1 and epsilon_2 be tuned? The experiment results report for a specific value of epsilons, without a clear intuition and/or guidance on choosing these two parameters.



**Summary Of The Paper:**

This paper consider the counterfactual generation problem; a counterfactual explanation of an input x with label y is a neighborhood point x' which is classified in the target class y'. This paper proposes a two step approach to generate counterfactual:

Step 1: take a gradient step to maximize the likelihood of class y' (conditional)
Step 2: take a gradient step to maximize the data likelihood (marginal)

The paper proposes to use sum-product network (SPN) to model the likelihood.

The numerical experiment shows that the counterfactual generated as such:
- requires few gradient evaluations,
- is visually appealing,
- is easy to compute,
- is effective.



**Summary Of The Review:**

The paper is an interesting read. I agree with the authors that this approach require few gradient evaluations, however, the paper lacks rigorous justification on (i) why the method is effective, (ii) whether it minimizes some neighborhood criteria, (iii) how to tune its parameters.

---

> ### Author Response · Authors · 2021-11-18
> **Response to Reviewer Hebe**
>
> Thank you for your comments! We are glad to address your questions.
>
> 1. The gradient in the first step is a first-order approximation of the model’s decision boundary near Xn, it points to the direction of desired class. However, the first gradient vector may not necessarily align with the manifold, which leads to a perturbation thats going off the manifold. Of course, a small enough perturbation could still keep the sample relatively close to the high-density region. However, epsilon_1 need to be big enough to cross the decision boundary to induce counterfactual class, which sometimes requires a big extrapolation from the query point. The example in figure 2 is a concrete case where the first step pertubation leads to an out-of-distribution image u. Please see the figure (figure 5) we added in the appendix for further illustration on this. On the 2D moon dataset, the first gradient step can extrapolate to low density region. Therefore the second step is needed.
> 2. S(x|y') is the unnormalized output, which means the increase of S(x|y') does not imply the decrease of S(x|y).  And the final prediction is determined by comparing S(x|y') and S(x|y), i.e. S(x|y') > S(x|y) yields prediction y' and y otherwise. In other words, S(x|y') - S(x|y) induces the decision boundary, and taking a gradient step towards the deepest ascent/decent of this quantity crosses the decision boundary. However, the direction for crossing decision boundary does not necessarily go through the high-density region, because the probability mass probably concentrates near regions that have a much smaller dimensionality than the original space where the data lives [1]. See again the 2D moon dataset in figure 5 as example.
> 3. epsilon_1 should at least be big enough to cross the decision boundary. In practice, one could start with small epsilon_1 and gradually increase epsilon_1 if the class prediction is not perturbed (for example when x and x’ are further apart) until the desired class is predicted. The experiments are also done with this practice, and it turns out to be very effective.
> 4. In the appendix, we mentioned how we tuned epsilon_1 and epsilon_2. In general, our approach is not very sensitive to these parameters, and especially epsilon_2 equals 1 works well enough throughout all the experiments we conducted, so we fixed epsilon_2 to 1 in the experiments. For epsilon_1, we start out with small epsilon_1 from 1 and gradually increase it (always by 10) if this is not big enough to perturb the prediction class. In practice, epsilon_1=1 are usually enough, epsilon_1=10 are sometimes also needed. But higher value for epsilon_1 or more tuning was not necessary for any experiment we conducted.
>
>
> [1] Representation Learning: A Review and New Perspectives. Yoshua Bengio, Aaron Courville, and Pascal Vincent. International Conference on Learning Representations.

---

### Decision · Program_Chairs · 2022-01-20

**Decision:**

Reject

**Comment:**

The only positive reviewer has not decided to step forward to champion the paper. All others have had a negative first impression which has not sufficiently changed after the answers from authors. My recommendation is based on the data that I have available: unfortunately for the authors the need of more clarity throughout and compelling results cannot be ignore/resolved with the info at hand.